# Endoparasitism of Golden Retrievers: Prevalence, risk factors, and associated clinicopathologic changes

**Elizabeth A. Kubas** [1]*, **Julie R. Fischer**[1], **Erin N. Hales**[2]

1 Department of Internal Medicine, Veterinary Specialty Hospital, San Diego, CA, United States of America,
2 Morris Animal Foundation, Denver, CO, United States of America

* elizabethkubas@gmail.com

**Data Availability Statement:** The data has has been made available through 2 different online data sites (https://data.world/ehales/grls-parasite-study and https://datacommons.morrisanimalfoundation. org/).

## Abstract

Endoparasitism is a common disease in dogs throughout their lifetime despite the widespread availability of inexpensive diagnostic tests and effective treatments. The consequences of host parasite interactions in otherwise apparently healthy dogs remains largely unknown. This cross-sectional study used complete blood count, serum biochemistry, and fecal flotation data collected from 3,018 young dogs (<3 years of age) enrolled within the Morris Animal Foundation Golden Retriever Lifetime Study (GRLS) to determine the prevalence of endoparasitism and compare bloodwork values of parasite positive and negative participants using logistic regression. Variables including age, gender, reproductive status, and geographic region at the time of evaluation were assessed to identify potential associations. To the authors' knowledge, a comprehensive assessment of clinicopathological changes associated with endoparasite infection in a large cohort has not been completed in the recent decade. The overall prevalence of endoparasitism was 6.99% (211/3018). Dogs who were parasite positive had statistically lower albumin ($P$ = 0.004), lower RBC count ($P$ = 0.01), higher neutrophil count ($P$ = 0.002), and higher platelet count ($P$ <0.001) as compared to parasite negative dogs. It was also concluded that dogs living in rural areas were more likely to have endoparasites than those living in suburban areas. Epidemiological data is crucial for the design and monitoring of prevention and control strategies. Identification of endoparasites by fecal testing is an essential tool to identify susceptible and resistant animals that can act as spreaders and reservoirs of intestinal parasites thereby enabling appropriate therapy and reducing the risk of new infection to animals and humans. Further epidemiological studies are needed to prevent, monitor, and develop new strategies to control endoparasites.

## Introduction

Endoparasites remain common in dogs throughout their lifetime despite the widespread availability of inexpensive diagnostic tests and effective treatments. Prevalence of infection with at least one intestinal parasite has been reported in up to 77.3% of dogs on post-mortem

**Funding:** The author(s) received no specific funding for this work.

**Competing interests:** The authors have declared that no competing interests exists.

**Abbreviations:** GRLS, Golden Retriever Lifetime Study; PLE, Protein losing enteropathy; CBC, Complete blood count.

examination in a 2017 study, many of which had no gastrointestinal signs at time of death [1]. The continuous fecal shedding of ova from undiagnosed or untreated dogs allows for the propagation of infection to other susceptible animals. Accurate knowledge of parasite prevalence and relevant risk factors are crucial for prompt diagnosis and treatment, as well as the design of optimal protocols for parasite control and owner education. Prevalence studies and assessment of risk factors for intestinal endoparasitism have been performed across various populations and geographical locations [2–10].

It is well established that endoparasitism can manifest in a variety of clinical presentations, ranging from asymptomatic to life-threatening. Clinicopathological changes associated with severe parasitism have been previously documented and include anemia, hypoalbuminemia, and eosinophilia [11–13]. It stands to reason that clinicopathologic changes are likely similarily variable and dependent on a variety of host and parasite factors. Information regarding the impact of subclinical infections is limited and, to the author's knowledge, a comprehensive assessment of clinicopathological changes associated with infection in a large cohort has not been completed in the recent decade.

Identifying additional risk factors and clinicopathological changes associated with endoparasitic infections would provide an additional tool for veterinarians to identify dogs that could benefit from additional fecal testing or empirical deworming. Prompt diagnosis and treatment benefits both the patient and the larger canine population as the infection would be treated preventing transmission to other dogs.

The objective of this study was to determine the prevalence of intestinal parasites within a large canine cohort and evaluate clinicopathologic data for any changes associated with endoparasite infections. We hypothesized that alterations in CBC and biochemical parameters exist due to the presence of a parasitic infection and its associated immune response. Additionally, we hypothesized that residing in more densely populated regions would be associated with a higher prevalence of endoparasitic infections due to increased contact between susceptible and infected dogs.

## Results

### Parasite status

At a baseline visit, both blood and fecal samples were collected for a complete blood count (CBC), serum biochemistry, and fecal floatation analysis. From this data two models were created using either the results from the CBC (S1 Data) or serum biochemistry panel (S2 Data). Of the total 3,018 dogs included in the CBC model, 211 (6.99%) were parasite positive and 2,807 (93.01%) were parasite negative for the. Of the 3,015 dogs included in the chemistry modeling, 211 (7.00%) were parasite positive and 2,804 (93.00%) were parasite negative (Table 3). Eleven dogs had both helminths and protozoa identified in their fecal samples. The remaining 2,807 fecal exams were considered negative having either no parasites observed or only non-pathogenic organisms. All dogs resided in the contiguous United States with a wide geographic distribution of positive dogs (Fig 1). The median age for dogs who were parasite positive was 11 months (minimum 5, maximum 27), whereas the parasite negative group was 14 months (minimum 4, maximum 38). There was no significant difference in overall parasite status based on age, sex, or reproductive status in either model (Tables 1 and 2).

### Complete blood cell count model

A complete blood cell model was created to evaluate the association between parasite status and the following parameters: absolute eosinophils, absolute neutrophils, absolute lymphocytes, absolute monocytes, platelet count, RBC, and hemoglobin (S1 Data). We found changes

## Parasite Positive Dogs Within the GRLS Cohort

• Positive    ·    Negative

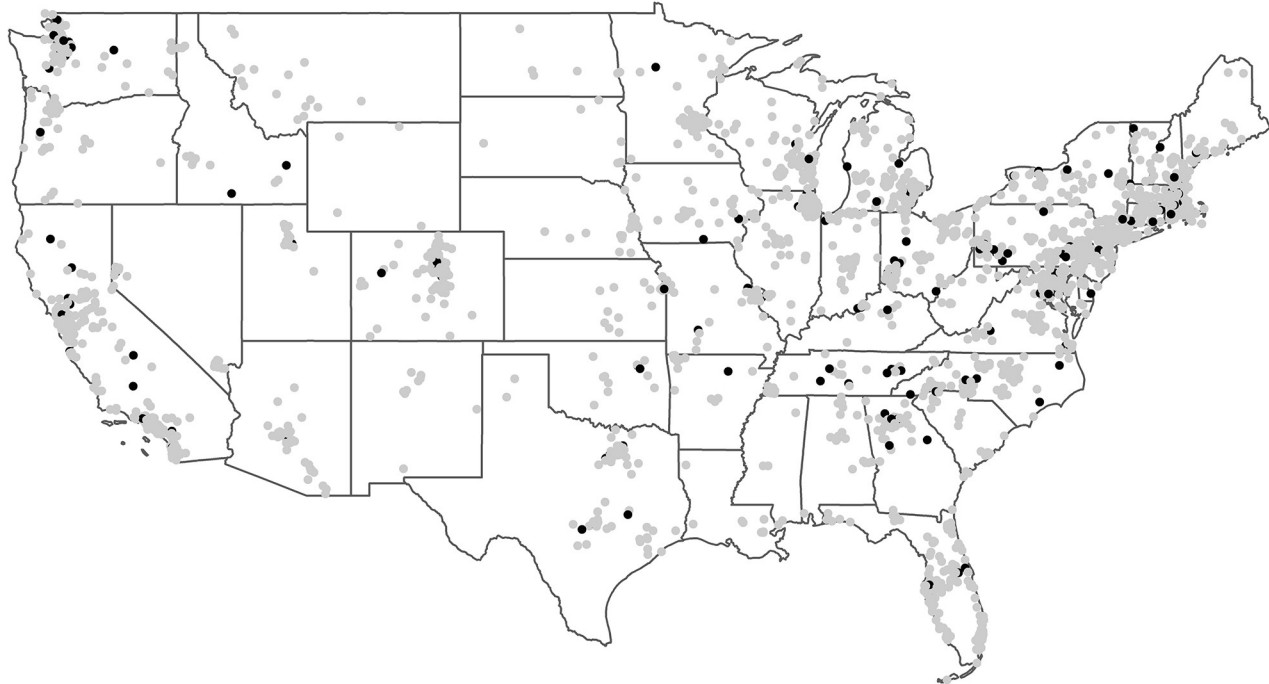

**Fig 1. Geographic locations for parasite positive and negative dogs within GRLS cohort.** Having a primary living location in the suburbs was associated with 0.57 decreased odds for being parasite positive (OR = 0.57, 95% CI = 0.42–0.77, P<0.001) as compared to having the primary residence in a rural area according to the CBC model. The chemistry model showed a similar decreased odds for a dog being parasite positive when living in the suburbs (OR = 0.54, 95% CI = 0.40–0.73, P<0.001).

in red blood cell count, absolute neutrophil, and platelet count that were associated with altered parasite status (Table 1). Specifically, every cell/μL increase in absolute neutrophils is associated with a 0.0001% (OR = 1.0001, 95% CI = 1.00004–1.0002, P = 0.002) increased odds of detecting a parasite via fecal centrifugation and floatation. Every 10,000 cells/μL increase in

**Table 1. Adjusted and unadjusted odds ratios and confidence intervals for the complete blood cell count model.** Bolded parameters were significant in the adjusted model.

| | Metrics | | Unadjusted | | Adjusted | |
|---|---|---|---|---|---|---|
| Variable | Positive | Negative | Odds Ratio (95%CI) | P value | Odds Ratio (95%CI) | P value |
| Male n(%) | 116 (3.84) | 1414 (46.85) | Reference | | | |
| Female n(%) | 95 (3.15) | 1393 (46.16) | 0.83 (0.63–1.10) | 0.20 | 0.75 (0.56–1.01) | 0.06 |
| Age (months) med. (min, max) | 11 (5, 27) | 14 (4, 38) | 0.97 (0.95–0.99) | 0.008** | 1.00 (0.97–1.02) | 0.97 |
| Intact n(%) | 146 (4.84) | 1604 (53.15) | Reference | | | |
| Neutered n(%) | 65 (2.15) | 1203 (39.86) | 0.59 (0.44–0.80) | <0.001*** | 0.92 (0.66–1.28) | 0.63 |
| Rural n(%) | 91 (3.02) | 792 (26.24) | Reference | | | |
| **Suburban n(%)** | **99 (3.28)** | **1720 (56.99)** | **0.50 (0.37–0.67)** | **<0.001***** | **0.57 (0.42–0.77)** | **<0.001***** |
| Urban n(%) | 21 (0.70) | 295 (9.77) | 0.62 (0.37–0.99) | 0.06 | 0.68 (0.40–1.11) | 0.14 |
| **Absolute Neutrophils med. (min, max)** | **5,992 (2,592, 16,287)** | **5,225 (1,599, 20,292)** | **1.0002 (1.0001–1.0002)** | **<0.001***** | **1.0001 (1.00004–1.0002)** | **0.002**** |
| **Platelets mean (±SD)** | **256.69 (69.99)** | **232.78 (64.19)** | **1.01 (1.003–1.008)** | **<0.001***** | **1.004 (1.002–1.006)** | **<0.001***** |
| **RBC mean (±SD)** | **6.30 (0.64)** | **6.40 (0.63)** | **0.64 (0.51–0.81)** | **<0.001***** | **0.72 (0.55–0.93)** | **0.01*** |

**Table 2. Adjusted and unadjusted odds ratios and confidence intervals for the chemistry model.** Bolded parameters remained significant in the adjusted model.

| | Metrics | | Unadjusted | | Adjusted | |
|---|---|---|---|---|---|---|
| Variable | Positive | Negative | Odds Ratio (95%CI) | P value | Odds Ratio (95%CI) | P value |
| Male n(%) | 116 (3.85) | 1411 (46.80) | Reference | | | |
| Female n(%) | 95 (3.15) | 1393 (46.20) | 0.83 (0.63–1.10) | 0.2 | 0.89 (0.66–1.18) | 0.41 |
| Age (months) med. (min, max) | 11 (5, 27) | 14 (4, 38) | 0.97 (0.95–0.99) | 0.01 | 1.00 (0.97–1.02) | 0.75 |
| Intact n(%) | 146 (4.84) | 1602 (53.13) | Reference | | | |
| Neutered n(%) | 65 (2.16) | 1202 (39.87) | 0.59 (0.44–0.80) | <0.001*** | 0.73 (0.53–1.00) | 0.05 |
| Rural n(%) | 91 (3.02) | 791 (26.24) | Reference | | | |
| **Suburban n(%)** | **99 (3.28)** | **1718 (56.98)** | **0.50 (0.37–0.67)** | **<0.001***** | **0.54 (0.40–0.73)** | **<0.001***** |
| Urban n(%) | 21 (0.70) | 295 (9.78) | 0.62 (0.37–0.99) | 0.06 | 0.66 (0.39–1.07) | 0.11 |
| **Albumin mean (±SD)** | **3.54 (0.26)** | **3.61 (0.22)** | **0.27 (0.15–0.49)** | **<0.001***** | **0.39 (0.21–0.75)** | **0.004**** |
| **Bilirubin med. (min, max)** | **0.2 (0.1, 0.4)** | **0.2 (0.1, 0.4)** | **0.005 (0.0004–0.05)** | **<0.001***** | **0.04 (0.003–0.47)** | **0.01*** |

**Table 3. Antech fecal results and corresponding parasite categories.** Numbers in parentheses denote the number of diagnoses.

| Parasite Positive | | Negative | |
|---|---|---|---|
| Helminths | Protozoa | | |
| Roundworms (59) | Coccidia (3) | None Seen (2744) | Sarcocystis (0) |
| Whipworms (35) | Giardia (79) | Nematodirus (0) | Capillaria (0) |
| Hookworms (28) | Isospora (23) | Strongyle (3) | Eimeria (69) |
| Tapeworms (2) | | Anoplocephala (0) | Rhabditiform (10) |
| Taeniid (0) | | Mites (2) | Trichomonad (1) |
| Alaria (1) | | Ascarids (1) | Moniezia (1) |

platelet count is associated with 0.004% (OR = 1.004, 95% CI = 1.002–1.0006, $P<0.001$) increased odds for a parasite positive fecal float. Every 10,000,000 cells/μl increase in red blood cells was associated with 0.72 decreased odds for being parasite positive (OR = 0.72, 95% CI = 0.55–0.93, $P = 0.01$).

## Chemistry model

A chemistry model was constructed to evaluate the association between the following parameters and parasite status: albumin, bilirubin, thyroxine (T4), glucose, sodium/potassium ratio, and total protein (S2 Data). Every g/dL increase in albumin is associated with 61% decreased odds (OR = 0.39, 95% CI = 0.21–0.75, $P = 0.004$) of having a positive fecal flotation (Fig 2). Every mg/dL increase in bilirubin is associated with 0.04 decreased odds (OR = 0.04, 95% CI = 0.003–0.47, $P = 0.01$) for a positive fecal result (Table 2).

## Discussion

Endoparasitism remains common in dogs in the United States despite frequent empiric treatment of puppies and routine fecal examinations. Within the analyzed group of young Golden Retrievers, we found the overall prevalence of endoparasite infection to be 6.99%. This is lower than many previously reported prevalence studies, which report up to 48.3% prevalence for any intestinal parasite [7]. Multiple statistically significant, albeit modest, clinicopathologic changes were identified between patients who were positive via fecal flotation for at least one pathogenic parasite when compared to patients with no evidence of endoparasitism.

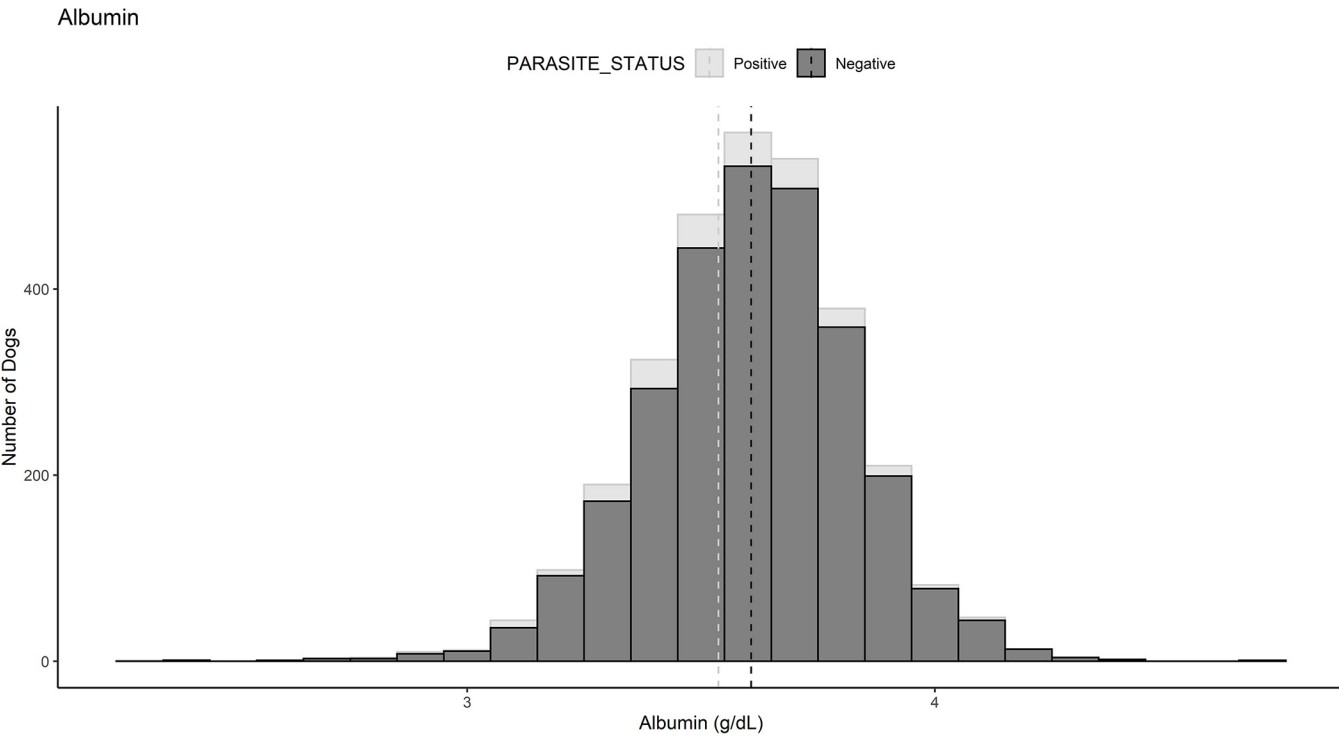

**Fig 2. Bar graph comparing serum albumin concentration (g/dL) between parasite positive and parasite negative cohorts.** Demonstrates increase in mean albumin concentrate associated with decreased odds (OR = 0.29, 95% CI = 0.21–0.75, P = 0.004) of parasite positive status. Dotted line indicates the mean for each group.

The reduced prevalence, as compared to earlier studies, may indicate increasing awareness of endoparasitism and its zoonotic potential as well as implementation of routine administration of broad spectrum anthelmintic. Prophylactic anthelmintic and use of preventative care have been shown to be increased in the GRLS population [14]. Importantly, as with previous prevalence studies, the prevalence of parasite infection is likely underestimated due to the inherent limited sensitivity of fecal flotation [1, 15]. Regardless, our data is in agreement with previous studies that endoparasitism remains common even in well cared for dogs [9, 10].

The CBC model for overall parasite status showed that a higher neutrophil count, higher platelet count, and lower RBC count were statistically associated with parasite positive dogs. While we cannot definitively say these changes are a direct result of the endoparasitism, we can offer explanations for these changes based on knowledge of the behavior of endoparasites and the pathology associated with infection. Previous studies have demonstrated helminths induce Th2 dominant immune responses, involving increased numbers of mucosal mast cell and intestinal eosinophils [16–19]. Activated eosinophils liberate helminthotoxic reactive oxygen species and granular proteins (including major basic protein) resulting in direct damage to parasites and host tissues. Eosinophils also modulate the immune response through cytokine and chemokine release. The neutrophilia and thrombocytosis seen in our data could be the result of cytokine stimulation of granulocyte-colony stimulating factor and thrombopoietin [20]. The mild thrombocytosis may be attributed to chronic intestinal hemorrhage. We postulate the lower RBC count could be due to multiple causes directly related to hemorrhage within the gastrointestinal tract, the decrease in serum iron from blood loss and/or the acute inflammatory response, or the direct effects of inflammation on the bone marrow [21, 22].

Future studies are need to identify if the small changes identified within our data are due to parasite infection, or causative of the infection itself.

Potential causes of the association seen with decreased albumin include blood loss (as discussed above) or development of a protein losing enteropathy (PLE) [22, 23]. Albumin's role as a negative acute phase protein may contribute to reduced serum albumin levels in response to chronic parasitic infection. Additionally, we cannot rule out the possibility that decreased albumin could have impacted the ability of the parasite to colonize an animal. Further investigation is needed to confirm our findings as the changes in odds ratio were small within our data. As previously stated, additional prospective studies may help provide further evidence that the changes identified are attributable to endoparasitism.

Though statistically significant, these clinicopathologic changes are subtle, particularly in comparison to the leukogram or biochemical changes seen with other infectious organisms. This may be due to parasitic immunomodulatory strategies or a generalized difference in the enteric immune response to chronic inflammation [18]. Additional studies would be beneficial to investigate these theories. Additionally, given the small magnitude of difference identified between infected or not infected dogs and lack of specificity of these changes for endoparasitic infection, it is clear that identification of clinicopathologic differences does not replace regular screening fecal testing and empiric treatment of high-risk patients.

Our analysis has demonstrated an increased likelihood of parasite infection associated with rural living locations as compared to suburban locations. This is in contrast to a previous study that showed patients residing in zip codes with higher population densities had a higher prevalence of intestinal parasitism [5]. The increased prevalence of endoparasitism in rural areas may be attributed to increased unsupervised outdoor time and roaming capabilities as well as potential differences in attitudes towards routine veterinary care in different geographical locations. Both these hypotheses require further investigation.

As a cross-sectional study, there were multiple inherent limitations. As previously discussed, a single fecal flotation was utilized to diagnose endoparasitism. The variable sensitivity of fecal flotation is very likely to underestimate true prevalence of infection. It has been demonstrated that fecal flotation fails to detect 5.6–93.7% of helminth infections, due to the inability to detect prepatent or single sex infections as well as false negatives because of intermittent parasite shedding or low intensity infections [1, 10]. Recent studies have demonstrated that utilizing fecal antigen tests in combination with conventional microscopy based fecal flotation greatly improve the diagnostic sensitivity for detection of endoparasite infection [1]. Thus, it is likely there were false negative results that may have limited our ability to detect the full magnitude of differences in biochemical and cell count changes between the two groups. Additionally, the study population was limited to patients under 3 years of age. This may have increased the overall prevalence as compared to a population that included all ages of dogs, because previous studies have shown increased prevalence of most parasite infections in younger dogs [2–10]. This choice was intentional to allow for comparison across a single time point as well as minimize the influence of concurrent disease often associated with an older population. It would be worthwhile repeating the study with mature adults to evaluate how our results compare to patients with fully developed immune systems. Lastly, while every effort was made to eliminate dogs with concurrent disease processes, given the study design, we cannot prove the lab work changes identified are attributable entirely to endoparasitism. An alternative, albeit considerably less likely explanation, would be that dogs with these preexisting clinicopathologic abnormalities have an increased susceptibility to endoparasitic infection. Additional follow up studies should be performed to confirm our findings and characterize the underlying molecular causes for the clinicopathologic changes seen in our study.

In conclusion, we found mild, but significant changes in serum biochemistry and CBC results including increases in a total neutrophil count, reduction in serum albumin, and elevated platelet counts in GRLS participant dogs with endoparasitism detected at the baseline time point. The only risk factor identified was a rural living environment as compared to a suburban one. The presence of this risk factor and clinicopathologic changes may heighten clinical suspicion of endoparasitism giving clinicians the opportunity to improve detection of subclinical infection despite flawed diagnostic tests.

## Materials and methods

### Case selection

This study was performed using data from dogs enrolled in the Golden Retriever Lifetime Study (GRLS) which enrolled a total of 3,044 dogs between 2012 to 2015. Enrollment required owners and veterinarians to complete an extensive questionnaire about the animal's basic health and environmental conditions. All dogs were between 4 months and 3 years of age and deemed free of life-threatening conditions by a veterinarian at the time of enrollment. At the baseline visit, whole blood, serum, and feces were collected by study veterinarians for CBC, serum biochemistry, T4, and fecal centrifuge and flotation. Dogs were removed from this study if they had no fecal at baseline (n = 19), had their fecal and serum collection performed at different times (n = 1) or had a gonadectomy date reported prior to date of birth (n = 6). Three participants lacked biochemistry results and therefore were only included in the CBC modeling cohort, resulting in a total of 3018 participants in the CBC modeling and 3015 participants in the chemistry modeling.

### Laboratory

All fecal samples were submitted to Antech Laboratory for fecal centrifuge and flotation (Memphis, TN). Fecal flotation results were used to classify each dog as parasite positive or negative (Table 3). A negative parasite read out included animals with no observed organisms, or only non-pathogenic organisms identified. Serum samples were also submitted to Antech Laboratory for CBC and biochemical analysis using routine methods.

### Statistical modeling

Logistical regression modeling was carried out using R version 3.6.1 [24] to investigate correlations between CBC or chemistry parameters and parasite status. CBC and chemistry parameters were tested using the appropriate test (t-test, Mann-Whitney U test, Chi-Square test, or Fisher's exact test) and selected for evaluation in the full model if $P<0.5$. Full models were fitted, then variables that did not achieve significance were removed. The final model was selected based on the lowest AIC score and likelihood ratio test indicating the least parameterize model was the best fit. All models were adjusted for age, sex, and reproductive status; male and intact were set as the reference. Age was determined using the dogs date of birth and the fecal flotation date as reported by the owner and lab respectively. Sex and reproductive status were determined using owner reported data. Residence area (rural, urban, or suburban) was collected via questionnaire from dog owners. Two models were built–a CBC model and a chemistry model. Within the CBC model, the association between parasite status and the following parameters were evaluated: absolute eosinophils, absolute neutrophils, absolute lymphocytes, absolute monocytes, platelet count, RBC, and hemoglobin. Within the chemistry model, we evaluated the association between the following parameters and parasite status: albumin, bilirubin, thyroxine (T4), glucose, sodium/potassium ratio, and total protein. Age,

sex, reproductive status, and residence type were included in both the complete blood cell count and chemistry models. A *P* value less than 0.05 was considered statistically significant for the full models.

## Supporting information

**S1 Data. Data used for the CBC model.**
(CSV)

**S2 Data. Data used for the Chemistry model.**
(CSV)

## Acknowledgments

The Golden Retriever Lifetime Study is funded by the Morris Animal Foundation. We would like to thank the owners, breeders, and primary care veterinarians for their invaluable contribution to the Golden Retriever Lifetime Study.

## Author Contributions

**Conceptualization:** Elizabeth A. Kubas.

**Data curation:** Erin N. Hales.

**Formal analysis:** Erin N. Hales.

**Software:** Erin N. Hales.

**Supervision:** Julie R. Fischer.

**Writing – original draft:** Elizabeth A. Kubas.

**Writing – review & editing:** Elizabeth A. Kubas, Julie R. Fischer, Erin N. Hales.

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
