## [Decision Letter · Decision Letter 0]

9 Aug 2021

PONE-D-21-20784

Intestinal parasitism of Golden Retrievers: prevalence, risk factors, and associated clinicopathologic changes

PLOS ONE

Dear Dr. Kubas,

Thank you for submitting your manuscript to PLOS ONE. After careful consideration, we feel that it has merit but does not fully meet PLOS ONE’s publication criteria as it currently stands. Therefore, we invite you to submit a revised version of the manuscript that addresses the points raised during the review process.

The reviewers pointed out some major concerns with this study as currently written. Reviewer 1 points out that fecal flotation is a limited evaluation tool and these limitations should be mentioned to help contextualize the findings presented. The statistical analyses need to be done properly and the conclusions should be supported by the analyses. The presentation of the tables should be clarified and the discussion should be refined and focused. Please address all of the concerns raised by the reviewers in a resubmission.

We look forward to receiving your revised manuscript.

Kind regards,

Adler R. Dillman, Ph.D.

Academic Editor

PLOS ONE

2. We have note from your ethics statement that the current study is a secondary analysis of data collected as a part of the Golden Retriever Lifetime Study. Please could you provide additional information in regarding the original ethics approval details for the Golden Retriever Lifetime Study.

Reviewers' comments:

Reviewer's Responses to Questions

**Comments to the Author**

1. Is the manuscript technically sound, and do the data support the conclusions?

Reviewer #1: No

Reviewer #2: Yes

2. Has the statistical analysis been performed appropriately and rigorously? 

Reviewer #1: No

Reviewer #2: Yes

3. Have the authors made all data underlying the findings in their manuscript fully available?

Reviewer #1: Yes

Reviewer #2: Yes

4. Is the manuscript presented in an intelligible fashion and written in standard English?

Reviewer #1: No

Reviewer #2: Yes

5. Review Comments to the Author

Reviewer #1: General comments

Thank you for inviting me to review this manuscript entitled “Intestinal parasitism of Golden Retrievers: prevalence, risk factors, and associated clinicopathologic changes.” The study reports fecal flotation results from 3,018 Golden Retriever dogs and then evaluates CBC and serum chemistry from dogs with parasites to gain insight into clinical pathology. While not entirely novel, the research does focus on Golden Retrievers and provides a comprehensive overview of parasitism in this population using a somewhat limited evaluation tool (fecal flotation). Reference should be made to fecal antigen testing in the discussion when the limitations of this approach are mentioned. The statistics should be revisited as both the OR and the significance calculations are off with nearly identical confidence intervals in different categories. An OR of 1 means there is no difference, so the conclusions are not supported. In addition, the manuscript would benefit from inclusion of a boarded parasitologist to ensure the background is complete and to correct errors and misstatements regarding veterinary parasites.

Specific comments

Will require copy editing for typographical errors, wording, and punctuation. Examples include:

Line 43-44: Remove tracked change and reword for clarity to read “…complex and result in a spectrum of clinical manifestations from asymptomatic infection to death.”

Line 46: Remove articles (“a”)

Line 48, omit comma

Reviewer stopped copy editing at line 49 but encourages the authors to do so prior to resubmission

Line 62: rephrase to read “…parasites remain common in dogs throughout…”

Line 66: Remove “infective” since most parasite ova (Ancylostoma, Toxocara, Trichuris, etc.) are not infective when shed, they must mature in the environment over a period of days to weeks.

Line 71-74, Rephrase for clarity; reduction of infections on human health does not make sense, and what is “opinion practice design”? Omit “limited” from line 74 as protocols would depend on accuracy and availability of information, not of limited information.

Line 79-80, the authors seem unaware of several additional, more recent publications? These should be included in the revision.

Sweet S et al. A 3-year retrospective analysis of canine intestinal parasites: fecal testing positivity by age, U.S. geographical region and reason for veterinary visit. Parasit Vectors. 2021 Mar 20;14(1):173.

Stafford K et al. Detection of gastrointestinal parasitism at recreational canine sites in the USA: the DOGPARCS study. Parasit Vectors. 2020 Jun 1;13(1):275.

Little SE, Johnson EM, Lewis D, Jaklitsch RP, Payton ME, Blagburn BL, Bowman DD, Moroff S, Tams T, Rich L, Aucoin D. Prevalence of intestinal parasites in pet dogs in the United States. Vet Parasitol. 2009 Dec 3;166(1-2):144-52.

Line 110-112: This conflicts with methods (Line 263) and the discussion (Line 238) which states that only dogs 6 months to 2 years or “under 2 years of age” were included in the study.

Line 112-113: The age analysis is not valid since only dogs 0.5 to 2 years of age (or perhaps 4 months to 3 + years?) were included in the study.

Table 2, 3: These tables are very confusing and the conclusions drawn are not valid. Females but not males are listed, and neutered but not intact, but then both suburban and urban are listed and not rural which was purportedly the highest risk category. Please recast. Also, consider drafting different tables for population characteristics and blood values to make it easier to follow. In general, and OR of 1 means no difference so I am not clear on why the authors claim significance for neutrophils and platelets. In short, it is not correct.

Line 121-125: The text discusses suburbs vs rural but the tables list suburban and urban?

Discussion: Much too long and unfocused. For the amount of data presented, a 3-4 paragraph discussion will suffice. Please condense. The reference for line 152 is incorrect – reference 13 is a paper from Canada.

Line 160-162: Mention fecal antigen tests (Reference 1) improve sensitivity of detection and support limited sensitivity of fecal flotation alone.

Table 1: This data belongs in results rather than methods. The term “helminths” is preferable to “worms” as the latter is vernacular. Scientific names must be italicized. Moniezia is not a parasite of dogs but instead a parasite of ruminants. Ascarids are likely also not a parasite of dogs as it would have been listed as “roundworms” if it were. More detail is needed on these species, they should be removed from “parasite positive” if they are not canine parasites, and the data reanalyzed.

Methods: No description is provided for how population density of dog residence (rural, suburban, etc.) was determined, or how age and sex were analyzed statistically.

References: The formatting is erratic with page numbers omitted from several. Close editing is required prior to resubmission.

Reviewer #2: I recommend publishing, improving the quality of the manuscript and discussion. Here you will find some of the recommendations for the article titled “Intestinal parasitism of Golden Retrievers: prevalence, risk factors, and associated clinicopathologic changes”

The study has important, relevant data that help explore future methods or alternatives to assess the impact of low parasite load (<4 eggs per gram) on feces, unfortunately, it is not well written, and the relevant data is not well discussed. I recommend supporting data with studies done to understand subclinical parasite effects with other models and populations: https://doi.org/10.3389/fimmu.2018.02975. Systemic Cytokine and Chemokine Profiles in Individuals With Schistosoma mansoni Infection and Low Parasite Burden.

6. PLOS authors have the option to publish the peer review history of their article (what does this mean?). If published, this will include your full peer review and any attached files.

Reviewer #1: No

Reviewer #2: No

---

## [Author Response · Author response to Decision Letter 0]

27 Sep 2021

Thank you for reading our manuscript and providing us the opportunity to revise and improve upon it. Your input was greatly appreciated. Each comment has been addressed below. 

Reviewer 1 Comments

Title

The title needs to be consistent in all the manuscripts: intestinal, gastrointestinal, or endoparasites. I will recommend changing to “endoparasites.”

Thank you for this suggestion, we have changed the manuscript accordingly.

Abstract

The study titled “Intestinal parasitism of Golden Retrievers: prevalence, risk factors, and associated clinicopathologic changes” is a retrospective study with a representative data analysis that could be used to compare with other dog populations in rural and urban areas. 

However, the assumption that “dogs living in rural areas were more likely to have endoparasites than those living in suburban areas” needs to be removed, considering that the study was not focused on this issue and there is not enough knowledge to suggest this in the abstract.

It seems there is some confusion here as this statement was a conclusion from our study and not an assumption made. The text has been rephrased to read: “It was also concluded that dogs living in rural areas were more likely to have endoparasites than those living in suburban areas.” (Line 34-36) The authors felt this was an important finding in their statistical models and therefore wanted to keep it in the abstract. We hope this re-wording reduces confusion and more accurately portrays the results.

Abstract.

Line 33-37 I recommend “the identification of endoparasites by fecal testing is an essential tool to identify susceptible and resistant animals that can act as spreaders and reservoirs of intestinal parasites thereby enabling appropriate therapy and reducing the risk of new infection to animals and humans. It is crucial information for further epidemiological studies and to prevent and monitor and develop new strategies to control.

Thank you for your input, we have incorperated it into the manuscript with minor grammatical adjustments (Line 39-42).

Author summary

Line 47-49. “Our data suggest that subtle changes may be seen on routine blood work submitted on young dogs that should prompt further investigation to diagnose or rule out endoparasitism” I recommend adding “and develop coprological serological or molecular tests to get stronger epidemiological information.”

Thank you for this suggestion. We have added it to the text (Lines 55-58).

Introduction

Line 62-63. The same for the title here, I recommend Endoparasites rather than Gastrointestinal.

Thank you, this was addressed throughout the manuscript.

Line 63-66. As the work uses coprological methods, I recommend citing references that have been used. These methods described are helpful in “availability of inexpensive diagnostic tests and effective treatments” for epidemiological records in developed and undeveloped countries such as “Peña Quistial et al., 2020. Prevalence and associated risk factors of Intestinal parasites in rural high-mountain communities of the Valle del Cauca—Colombia

Thank you for your comment. The coprological methods referenced in our paper are basic and discussed in most of the previously cited references. With the reewrite, we have tried to focus our references on US work as there was some confusion surrounding literature from other countries brought to light during the review process. 

Line 63-66. I recommend highlighting how using complementary techniques can be helpful too. However, it is not necessary to arrive at a post mortem examination to get a better diagnosis as given the idea following ….” Prevalence of infection with at least one intestinal parasite has been reported in up to 77.3% of dogs on post-mortem examination in a 2017 study, many of which had no gastrointestinal signs at the time of death[1]”.

Thank you, we agree that post-mortem is not a clinically relevant diagnostic test, however this statistic highlights the poor senstivity of antemortem testing and the fact that we should not rely on gastrointestinal symptoms to have clinical suspicion for endoparasitism. The utility of complementary techniques such as fecal antigen tests were added to the discussion (lines 276-279). 

Line 66-67. In this phrase, authors need to be careful “The continuous fecal shedding of infective ova from undiagnosed or untreated dogs allows for the propagation of parasitic infection to other susceptible animals.” There are two kinds of dog populations: susceptible and resistant populations; all of them spread parasites. The difference is that resistance is asymptomatic and silently keeps parasites while susceptible will develop symptoms, and both will spread parasites to other animals’ populations, both susceptible and resistant again. I suggest a reference that can support this concept much better. 

Thank you for your comment. While the authors understand the complexity, they would not consider an asymptomatic patient to be resistant to infection if they are harboring and shedding parasites. This patient could become symptomatic at any time (i.e. become immune compromised from endocrine disease or underlying neoplasia). 

Line 68-71. I recommend talking that there is an enormous number of zoonotic parasites, protozoa and helminth that give a better idea of the risk for humans and animals “Furthermore, the zoonotic risk of several parasites adds further importance to the prompt identification of parasitic infections. An important example of this is Ancylostoma spp. and Toxocara spp., which cause cutaneous larva migrans and visceral or ocular larval migrans respectively [2]”.

We completely agree that there are an enormous number of zoonotic parasites that are relevant to human health, but this aspect was removed from the manuscript to improve the focus/clarity. 

Line 71-72 The author arrives at One Health Approach suddenly without context as the solution to reduce asymptomatic infections, so this idea needs to be introduced in a better way “Reduction of asymptomatic infections on both companion animals, and human health can be achieved through a one health approach”.

We agree with your comment, the introduction was heavily revised for clarity. 

Line 79-80 “animals residing across the United States was published in 2009 [8], but there have been no updated studies in the most recent decade”. This study updates the information in the USA after a decade, so I think that needs to be in the abstract.

Upon further review of literature, there have been more recent prevalence studies performed in the United States. Therefore this was not added to the abstract. Thank you for the suggestion that made us look deeper into the literature.

Sweet S et al. A 3-year retrospective analysis of canine intestinal parasites: fecal testing positivity by age, U.S. geographical region and reason for veterinary visit. Parasit Vectors. 2021 Mar 20;14(1):173.

Stafford K et al. Detection of gastrointestinal parasitism at recreational canine sites in the USA: the DOGPARCS study. Parasit Vectors. 2020 Jun 1;13(1):275.

Line 84-89. The way as has been written is not clear first studies support severe parasitism and in the absence of gastrointestinal signs…and after clinicopathological changes associated with low infection level in asymptomatic animals has not been documented (maybe could be improved, please clarify) “Several clinicopathological changes associated with severe parasitism have been documented and include anemia, hypoalbuminemia, and eosinophilia [9-12]. Previously published literature has also reported that endoparasitic infections can be present in the absence of gastrointestinal signs [2-5]. However, information regarding the impact on animal well-being is limited, and a comprehensive assessment of clinicopathological changes associated with infection levels in a large cohort has not been completed in the recent decade.

Thank you for this comment – the text has been clarified below. (Lines 93-100)

 “It is well established that endoparistism can manifest in a variety of clinical presentations, ranging from asymptomatic to life-threatening. Clinicopathological changes associated with severe parasitism have been previously documented and include anemia, hypoalbuminemia, and eosinophilia [9-12]. It stands to reason that clinicopathologic changes are likely similarily variable and dependent on a variety of host and parasite factors. Information regarding the impact of clinically inapparent infections is limited, and to the author’s knowledge a comprehensive assessment of clinicopathological changes associated with infection in a large cohort has not been completed in the recent decade.”

Line 90-95. Here the idea is that preventive medicine needs to be applied based on evidence, and parasite deworming needs to be improved, detecting asymptomatic animals, reducing zoonotic and animal transmission. (Please improve the writing) “Identifying risk factors and clinicopathological changes associated with endoparasite infections would provide an additional clinical tool for veterinarians in their assessment of dogs for additional fecal testing or empirical deworming. This has the opportunity to benefit both the patient and the larger canine population as the infection would be treated, preventing transmission to other dogs. For parasites of zoonotic importance, human transmission risk would also be reduced”.

Thank you for your comment, the manuscript has been reworded for clarity. It now reads (Lines 112-117): 

Identifying additional risk factors and clinicopathological changes associated with endoparasitic infections would provide an additional tool for veterinarians to identify dogs that could benefit from additional fecal testing or empirical deworming. Prompt diagnosis and treatment benefits both the patient and the larger canine population as the infection would be treated preventing transmission to other dogs

Line 96 -101. Here in summary, the authors did a retrospective study or a cohort study associating clinical data with endoparasite infection, but how the study let to achieve this hypothesis is not clear as well as how the study tests the density of the populations as a risk factor to spread parasites is not clear? The objective of this study was to determine the prevalence of intestinal parasites within a large canine cohort and evaluate clinicopathologic data for any changes associated with endoparasite infections. We hypothesized that alterations in CBC and biochemical parameters exist due to the presence of a parasitic infection and its associated immune response. Additionally, we hypothesized that residing in more densely populated regions may increase the likelihood of endoparasites infection.

Thank you for your comment – we compared prevalence of parasites in dogs that are identified to live in rural, suburban, or urban locations. This information was available from the questionaire the owners submitted as a part of being enrolled in the Golden Retriever Lifetime Study. Therefore, we were hypothesized that the dogs living in an urban setting would have an increased chance of being parasite positive rather than dogs in rural settings who have less contact with other dogs. 

Materials and Methods

Line 259 case selection. The way the cases were selected is clear and the information available.

Line 273 Laboratory. 

Line 280 Statistical modeling is appropriately described

Results. Results are correctly described

Discussion.

Line 147-148 “Endoparasitism remains common in dogs in the United States despite frequent empiric treatment of puppies and routine fecal examinations”. So why is the treatment of puppies in the USA empiric, there is no diagnosis or treatment based on evidence? So it is possible to be more precise.

Thank you for your comment, you are correct. In my experience as a small animal clinician, all puppies are treated empirically for endoparasites during their first visit at 6-8 weeks of age, most often without a fecal float. Subsequently treatment is often based on diagnostic results, but for owners who decline diagnostics, treatment can continue to be empiric. It is not ideal, but veterinary medicine is often a compromise between what is best for a patient/client and gold standard medicine.

Line 153-155. Here, the authors need to clarify that it is not in Canada or Australia studies in this study. Is this correct?. Multiple statistically significant, albeit modest, clinicopathologic changes were identified between patients who were positive via fecal flotation for at least one pathogenic parasite when compared to patients with no evidence of endoparasitism.

Thank you for your comment, I have removed the Canadian and Australian studies as they are less relevant to our data and may confuse the reader. 

Line 157-162. In the discussion, there is no emphasis in the results. It is not consistent whether the reduced prevalence is the effect of medical care and people conscious of zoonotic risk factors or the sum of all of them and the broad-spectrum parasite. After that, minimize the test saying that the data is not so true prevalence. I will say that the test has limited sensitivity but still is available to see the iceberg. So please be more consistent about what you want to say. “The reduced prevalence may indicate increasing awareness of endoparasitism and its zoonotic potential as well as the implementation of routine administration of broad-spectrum anthelmintics. Prophylactic anthelmintic and the use of preventative care are increased in the GRLS population [15]. Additionally, as with previous prevalence studies, the true prevalence of parasite infection is likely higher than reported values due to the limited sensitivity of fecal flotation [16]”.

Thank you for these comments. You make an excellent point, many factors have affected our prevalence data, but unfortunately from our data and analysis we cannot be definitive about which factors had the most influence. I agree that while the sensitivity of fecal float is not perfect, it does provide useful epidemiological information. This is particularly relevant because the vast majority of veterinarians in the United States are relying on fecal flotation for diagnosis of parasitic infections. I do not think recognizing the limitations of fecal flotation detracts from the validity of our data, but I have added additional comments that reiterate why the data is still important and relevant. 

Lines 190-192: Regardless, our data is in agreement with previous studies that endoparasitism remains common even in well cared for dogs [9,10].

Line 164-174 Some arguments in the document could be much better discussed and supported in the literature, such as the Review “The immune response to parasitic helminths of veterinary importance and its potential manipulation for future vaccine control strategies. Neil Foster & Hany M. Elsheikha” DOI 10.1007/s00436-012-2832-y 

Thank you for this reference. I have reviewed it and clarified the discussion where appropriate, though I am afraid a more detailed review of host-parasite interactions is beyond the scope of this manuscript. 

Lines 198-204: Previous studies have demonstrated helminths induce Th2 dominant immune responses, involving increased numbers of mucosal mast cell and intestinal eosinophils [17-19]. Activated eosinophils liberate helminthotoxic reactive oxygen species and granular proteins (including major basic protein) resulting in direct damage to parasites and host tissues. Eosinophils also modulate the immune response through cytokine and chemokine release. The neutrophilia and thrombocytosis seen in our data are likely the result of cytokine stimulation of granulocyte-colony stimulating factor and thrombopoietin.

Conclusions

Are correctly address

Reviewer 2 Comments:

Will require copy editing for typographical errors, wording, and punctuation. Examples include:

Line 43-44: Remove tracked change and reword for clarity to read “…complex and result in a spectrum of clinical manifestations from asymptomatic infection to death.”

Line 46: Remove articles (“a”)

Line 48, omit comma

Reviewer stopped copy editing at line 49 but encourages the authors to do so prior to resubmission

Thank you for your recommendations, the manuscript was heavily edited for clarity, grammar, and typography prior to resubmission.

Line 68: rephrase to read “…parasites remain common in dogs throughout…”

Thank you for the suggestion. The sentence was rephrased to read “Endoparasites remain common in dogs throughout their lifetime despite the widespread availability of inexpensive diagnostic tests and effective treatments.” (Lines 75-77)

Line 66: Remove “infective” since most parasite ova (Ancylostoma, Toxocara, Trichuris, etc.) are not infective when shed, they must mature in the environment over a period of days to weeks.

You are correct, the phrasing was adjusted for clarity and accuracy as suggested. 

Line 75 “The continuous fecal shedding of ova from undiagnosed or untreated dogs allows for the propagation of infection to other susceptible animals.”

Line 71-74, Rephrase for clarity; reduction of infections on human health does not make sense, and what is “opinion practice design”? Omit “limited” from line 74 as protocols would depend on accuracy and availability of information, not of limited information.

Thank you for the input. The authors agree this was confusing and have rephrased to: “Accurate knowledge of parasite prevalence and relevant risk factors are crucial for prompt diagnosis and treatment, as well as the design of optimal protocols for parasite control and owner education.” (Lines 80-84)

Line 79-80, the authors seem unaware of several additional, more recent publications? These should be included in the revision.

Sweet S et al. A 3-year retrospective analysis of canine intestinal parasites: fecal testing positivity by age, U.S. geographical region and reason for veterinary visit. Parasit Vectors. 2021 Mar 20;14(1):173.

Stafford K et al. Detection of gastrointestinal parasitism at recreational canine sites in the USA: the DOGPARCS study. Parasit Vectors. 2020 Jun 1;13(1):275.

Little SE, Johnson EM, Lewis D, Jaklitsch RP, Payton ME, Blagburn BL, Bowman DD, Moroff S, Tams T, Rich L, Aucoin D. Prevalence of intestinal parasites in pet dogs in the United States. Vet Parasitol. 2009 Dec 3;166(1-2):144-52.

Thank you for bringing this to our attention, these sources have been reviewed and added to the manuscript where relevant. 

Line 110-112: This conflicts with methods (Line 263) and the discussion (Line 238) which states that only dogs 6 months to 2 years or “under 2 years of age” were included in the study.

The authors agree this was confusing. The text in the methods has been updated to read “…between 4 months and 3 years of age…” (line 315)

Line 112-113: The age analysis is not valid since only dogs 0.5 to 2 years of age (or perhaps 4 months to 3 + years?) were included in the study.

Although it was not significant, age in months was included in the modeling since it has previously been associated with parasite status. We have updated the methods to indicate that dogs up to their 3-year-old year are included in the analysis (Line 311). We feel it is important to show that this variable was indeed not significant in the current analysis.

Table 2, 3: These tables are very confusing and the conclusions drawn are not valid. Females but not males are listed, and neutered but not intact, but then both suburban and urban are listed and not rural which was purportedly the highest risk category. Please recast. Also, consider drafting different tables for population characteristics and blood values to make it easier to follow. In general, and OR of 1 means no difference so I am not clear on why the authors claim significance for neutrophils and platelets. In short, it is not correct.

The authors appreciate that the tables caused confusion and therefore have updated them to include the reference categories (rural, male, and intact). The population characteristics were included within the models and therefore the authors feel it is important to report all the numbers together. We acknowledge that the OR is very close to 1 for platelets and neutrophils, however the modeling indicated they were highly significant, based on p-value, even with the very small odds ratios. This means there is a small, albeit significant difference which is therefore important to discuss. 

Line 121-125: The text discusses suburbs vs rural but the tables list suburban and urban?

The authors recognize the tables cause confusion and have recast them. Rural was set as the reference category for the logistical regression modeling, leading to the results mentioned in this comment. The tables have been updated to include the reference category for clarity (Lines 141 and 142).

Discussion: Much too long and unfocused. For the amount of data presented, a 3-4 paragraph discussion will suffice. Please condense. The reference for line 152 is incorrect – reference 13 is a paper from Canada.

Thank for your comments, we agree the discussion could be consolidated and clarified. The discussion was heavily edited to remove redundant or less relevant information. Line 152 was deleted in the restructuring, but we apologize that the reference cited was incorrect. All references have been checked for accuracy in the editing process as well. 

Line 160-162: Mention fecal antigen tests (Reference 1) improve sensitivity of detection and support limited sensitivity of fecal flotation alone.

The limitations of fecal flotation and more recent publications demonstrating the utility of fecal antigen tests in addition to fecal flotation were added. Thank you for this suggestion.

Line 274-279: “It has been demonstrated that fecal flotation fails to detect 5.6-93.7% of helminth infections, due to the inability to detect prepatent or single sex infections as well as false negatives because of intermittent parasite shedding or low intensity infections. Recent studies have demonstrated that utilizing fecal antigen tests in combination with conventional microscopy based fecal flotation greatly improve the diagnostic sensitivity for detection of endoparasite infection.” 

Table 1: This data belongs in results rather than methods. The term “helminths” is preferable to “worms” as the latter is vernacular. Scientific names must be italicized. Moniezia is not a parasite of dogs but instead a parasite of ruminants. Ascarids are likely also not a parasite of dogs as it would have been listed as “roundworms” if it were. More detail is needed on these species, they should be removed from “parasite positive” if they are not canine parasites, and the data reanalyzed.

The table heading has been updated with helminths instead of worms and the scientific names have been italicized. Thank you for these suggestions.

Additionally, the two dogs with Moniezia and unspecified Ascarids have been re-classified and the models re-evaluated. This led to eosinophils no longer needed to be retained in the model and updated estimates for odds ratios and p-values. All numbers throughout the text have been checked and updated.

We opted to leave the table in the methods as it describes how the dogs were categorized into the parasite positive or negative groups. However, we have added a reference to the table within the results section. 

Methods: No description is provided for how population density of dog residence (rural, suburban, etc.) was determined, or how age and sex were analyzed statistically.

This was an oversight on the part of the authors and we appreciate the opportunity to add additional information. Clarification has been added to the methods section to indicate all population characteristics noted in this comment were analyzed within both logistical regression models. The added sentences are lines 333-336 and 342-343. The added text is listed below:

Age was determined using the dogs date of birth and the fecal flotation date as reported by the owner and lab respectively. Sex and reproductive status were determined using owner reported data. Residence area (rural, urban, or suburban) was collected via questionnaire from dog owners.

Age, sex, reproductive status, and residence type were included in both the complete blood cell count and chemistry models.

References: The formatting is erratic with page numbers omitted from several. Close editing is required prior to resubmission.

The formatting of references was fixed, we apologize for the oversight. Some references did not have page numbers readily available (even after searching other publications that also used the same reference). The page numbers were removed from all journal articles for the sake of uniformity.

---

## [Decision Letter · Decision Letter 1]

15 Nov 2021

PONE-D-21-20784R1Endoparasitism of Golden Retrievers: prevalence, risk factors, and associated clinicopathologic changesPLOS ONE

Dear Dr. Kubas,

Thank you for submitting your manuscript to PLOS ONE. After careful consideration, we feel that it has merit but does not fully meet PLOS ONE’s publication criteria as it currently stands. Therefore, we invite you to submit a revised version of the manuscript that addresses the points raised during the review process.

Thank you for addressing previous reviewer comments in this revision. There are just a few additional suggestions that have been made that would improve the manuscript, including providing access to the raw data.

We look forward to receiving your revised manuscript.

Kind regards,

Adler R. Dillman, Ph.D.

Academic Editor

PLOS ONE

Journal Requirements:

Additional Editor Comments (if provided):

Thank you for submitting this revised version, and for addressing previous reviewer comments. There are just a few additional suggestions that have been made that would improve the manuscript, including providing access to the raw data.

Reviewers' comments:

Reviewer's Responses to Questions

**Comments to the Author**

1. If the authors have adequately addressed your comments raised in a previous round of review and you feel that this manuscript is now acceptable for publication, you may indicate that here to bypass the “Comments to the Author” section, enter your conflict of interest statement in the “Confidential to Editor” section, and submit your "Accept" recommendation.

Reviewer #2: All comments have been addressed

Reviewer #3: (No Response)

2. Is the manuscript technically sound, and do the data support the conclusions?

Reviewer #2: Yes

Reviewer #3: Partly

3. Has the statistical analysis been performed appropriately and rigorously? 

Reviewer #2: Yes

Reviewer #3: Yes

4. Have the authors made all data underlying the findings in their manuscript fully available?

Reviewer #2: Yes

Reviewer #3: No

5. Is the manuscript presented in an intelligible fashion and written in standard English?

Reviewer #2: Yes

Reviewer #3: Yes

6. Review Comments to the Author

Reviewer #2: (No Response)

Reviewer #3: Thank you for submitting this interesting and well written study. I have a few suggestions which relate to communicating and interpreting the study findings (below). I also am not sure that the data is considered "fully available" as per PLOS ONE's data policy. An email is provided but it would be preferable to upload the data to a repository such as on github.com where it can be accessed directly.

1. The authors write that in future lab tests could be used to prioritize screening for GI parasites. However, I think they need to address the diagnostic value of the lab tests, and not just the statistical significance of the associations between lab tests and endoparasitism. It is clear from the data they provide that the magnitude of the effects are very small and there would be a large overlap in healthy and infected dogs. Also, they state a large proportion of cases may go undetected. Might it be more advisable to suggest screening and/or treatment of all young dogs?

2. The authors infer that the parasites cause the lab test changes, but do not address alternative explanations that cannot be ruled out from this cross-sectional study design i.e. reverse causality or confounding. I think it would improve the manuscript if this was highlighted in the discussion.

3. I think the limited age range of the dogs included in the study is important and should be mentioned early on in the manuscript - in the abstract, at least.

4. Likewise, the fact that the outcome and risk factor variables were assessed at the same appointment time is important and should be mentioned in the abstract and early in the paper. It could be argued that it's a cross-sectional study and not a retrospective study, as the study participants are not followed forward through time (as in a retrospective cohort study) or assessed "back through time" as in a retrospective case control study.

4. The full logistic regression models were fit and then parameters were dropped based on AIC and likelihood ratio tests. Firstly, which of these two metrics did you use and/or how did you combine them? Also, how did you select these initial parameters for the model - since this is a very large study, I assume that other data was collected at the same time?

5. In the results you refer to the "chemistry model" and the "blood cell model" - I think since the results come before the methods it would be helpful to give more precise explanations.

7. PLOS authors have the option to publish the peer review history of their article (what does this mean?). If published, this will include your full peer review and any attached files.

Reviewer #2: No

Reviewer #3: **Yes: **Wendy Beauvais

---

## [Author Response · Author response to Decision Letter 1]

22 Dec 2021

Response to editors

- No changes to the works cited page were made during this revision

- We have provided our raw data (see below)

Response to reviewers

- All data needs to be made available 

o Thank you for your comment – The data has been added as supplementary material to the manuscript and made available through 2 different online data sites (https://data.world/ehales/grls-parasite-study and https://datacommons.morrisanimalfoundation.org/).

- 1. The authors write that in future lab tests could be used to prioritize screening for GI parasites. However, I think they need to address the diagnostic value of the lab tests, and not just the statistical significance of the associations between lab tests and endoparasitism. It is clear from the data they provide that the magnitude of the effects are very small and there would be a large overlap in healthy and infected dogs. Also, they state a large proportion of cases may go undetected. Might it be more advisable to suggest screening and/or treatment of all young dogs?

o Thank you for your comment, we agree and have changed the manuscript accordingly. 

Line 186 we have added:

• Additionally, given the small magnitude of difference identified between infected or not infected dogs and lack of specificity of these changes for endoparasitic infection, it is clear that identification of clinicopathologic differences does not replace regular screening fecal testing and empiric treatment of high-risk patients. 

- 2. The authors infer that the parasites cause the lab test changes, but do not address alternative explanations that cannot be ruled out from this cross-sectional study design i.e. reverse causality or confounding. I think it would improve the manuscript if this was highlighted in the discussion.

o Thank you for your comment, we agree and have changed the manuscript in several location including:

Line 56 reads:

• Our data suggest that subtle changes may be seen on routine bloodwork submitted on young dogs and those may be associated with a positive fecal float result.

Line 200 reads:

• Additionally, we cannot rule out the possibility that decreased albumin could have impacted the ability of the parasite to colonize an animal.

Line 255-258 now read:

• Lastly, while every effort was made to eliminate dogs with concurrent disease processes, given the study design, we cannot prove the lab work changes identified are attributable entirely to endoparasitism. Additional follow up studies should be performed to confirm our findings and characterize the underlying molecular causes for the clinicopathologic changes seen in our study.

Lines 246-248 added:

• An alternative, albeit considerably less likely explanation, would be that dogs with these preexisting clinicopathologic abnormalities have an increased susceptibility to endoparastitic infection.

Lines 180-183 added:

• While we cannot definitively know these changes are a direct result of the endoparasitism, we can offer explanations for these changes based on knowledge of the behavior of endoparasites and the pathology associated with infection.

- 3. I think the limited age range of the dogs included in the study is important and should be mentioned early on in the manuscript - in the abstract, at least.

o Thank you – we have added the age of the population in the abstract (line 25).

Line 23 – 28 now reads

This cross sectional study used complete blood count, serum biochemistry, and fecal flotation data collected from 3,018 young dogs (<3 years of age) enrolled within the Morris Animal Foundation Golden Retriever Lifetime Study (GRLS) to determine the prevalence of endoparasitism and compare bloodwork values of parasite positive and negative participants using logistic regression.

- 4. Likewise, the fact that the outcome and risk factor variables were assessed at the same appointment time is important and should be mentioned in the abstract and early in the paper. It could be argued that it's a cross-sectional study and not a retrospective study, as the study participants are not followed forward through time (as in a retrospective cohort study) or assessed "back through time" as in a retrospective case control study.

o Thank you for your comment, I see what you are saying. Line 28 I have added:

Variables including age, gender, reproductive status, and geographic location at the time of initial evaluation were assessed to identify potential risk factors

o We have also added a line to the beginning of the results to expressly state that all samples were collected at one time point.

Line 111-114: At a baseline visit, both blood and fecal samples were collected for a complete blood count (CBC), serum biochemistry, and fecal floatation analysis. From this data two models were created using either the results from the CBC (supplementary material 1) or serum biochemistry panel (supplementary material 2).

o The authors agree that this is more of a cross sectional design and have altered the manuscript to reflect this. Major changes to the discussion are listed below with line numbers.

Line 193-195: Future studies are need to identify if the small changes identified within our data are due to parasite infection, or causative of the infection itself.

Line 199-201: As previously stated, additional prospective studies may help provide further evidence that the changes identified are attributable to endoparasitism.

- 4. The full logistic regression models were fit and then parameters were dropped based on AIC and likelihood ratio tests. Firstly, which of these two metrics did you use and/or how did you combine them? Also, how did you select these initial parameters for the model - since this is a very large study, I assume that other data was collected at the same time?

o The authors agree this is important information and have added it to the methods section of the manuscript, Lines 299-304.

CBC and chemistry parameters were tested using the appropriate test (t-test, Mann-Whitney U test, Chi-Square test, or Fisher’s exact test) and selected for evaluation in the full model if P<0.5. Full models were fitted, then variables that did not achieve significance were removed. The final model was selected based on the lowest AIC score and likelihood ratio test indicating the least parameterize model was the best fit. 

- 5. In the results you refer to the "chemistry model" and the "blood cell model" - I think since the results come before the methods it would be helpful to give more precise explanations.

o Thank you for your comment, we agree and have added an explanation on the first line of each results paragraph as well as a line at the beginning of the results section.

Line 111-114: At a baseline visit, both blood and fecal samples were collected for a complete blood count (CBC), serum biochemistry, and fecal floatation analysis. From this data two models were created using either the results from the CBC (supplementary material 1) or serum biochemistry panel (supplementary material 2).

Line 138-140: A complete blood cell model was created to evaluate the association between parasite status and the following parameters: absolute eosinophils, absolute neutrophils, absolute lymphocytes, absolute monocytes, platelet count, RBC, and hemoglobin.

Line 150-152: A chemistry model was constructed to evaluate the association between the following parameters and parasite status: albumin, bilirubin, thyroxine (T4), glucose, sodium/potassium ratio, and total protein.

---

## [Editor Report · Decision Letter 2]

21 Jan 2022

Endoparasitism of Golden Retrievers: prevalence, risk factors, and associated clinicopathologic changes

PONE-D-21-20784R2

Dear Dr. Kubas,

We’re pleased to inform you that your manuscript has been judged scientifically suitable for publication and will be formally accepted for publication once it meets all outstanding technical requirements.

Kind regards,

Adler R. Dillman, Ph.D.

Academic Editor

PLOS ONE

Additional Editor Comments (optional):

Thank you for addressing the additional suggested revisions.
---

## [Editor Report · Acceptance letter]

10 Feb 2022

PONE-D-21-20784R2 

Endoparasitism of Golden Retrievers: prevalence, risk factors, and associated clinicopathologic changes 

Dear Dr. Kubas:

I'm pleased to inform you that your manuscript has been deemed suitable for publication in PLOS ONE. Congratulations! Your manuscript is now with our production department. 

Kind regards, 

on behalf of

Dr. Adler R. Dillman 

Academic Editor

PLOS ONE